# The Complex Interplay between Vaginal Microbiota, HPV Infection, and Immunological Microenvironment in Cervical Intraepithelial Neoplasia: A Literature Review

**DOI:** 10.3390/ijms23137174

**Published:** 2022-06-28

**Authors:** Barbara Gardella, Marianna Francesca Pasquali, Marco La Verde, Stefano Cianci, Marco Torella, Mattia Dominoni

**Affiliations:** 1Department of Obstetrics and Gynecology, IRCCS Foundation Policlinico San Matteo, 27100 Pavia, Italy; barbara.gardella@gmail.com (B.G.); matti.domino@gmail.com (M.D.); 2Department of Clinical, Surgical, Diagnostic and Paediatric Sciences, University of Pavia, 27100 Pavia, Italy; 3Obstetrics and Gynecology Unit, Department of Woman, Child and General and Specialized Surgery, University of Campania, Luigi Vanvitelli, 80138 Naples, Italy; marco.laverde88@gmail.com (M.L.V.); marcotorella@iol.it (M.T.); 4Department of Obstetrics and Gynecology, University of Messina, 98122 Messina, Italy; stefanoc85@hotmail.it

**Keywords:** microbiota, CIN, HPV, inflammation

## Abstract

Background: in recent years, many studies were carried out to explore the role of vaginal microbiota in HPV infections and cervical intraepithelial neoplasia (CIN) progression. The aim of this study was to conduct a review of the literature to analyze the interaction between the vaginal microbiota, the CIN, and the immunological response. Methods: we performed a literature search, considering papers published between November 2015 and September 2021. Results: despite significant evidence suggesting a role of vaginal microbiota in the pathogenesis of HPV-related lesions, some studies still struggle to demonstrate this correlation. However, the vaginal microbiota of HPV-positive women shows an increased diversity, combined with a reduced relative abundance of *Lactobacillus* spp. and a higher pH. In cervical dysplasia progression, a strong association is found with new bacteria, and with the deregulation of pathways and hyperexpression of cytokines leading to chronic inflammation. Conclusions: in HPV progression, there is a strong correlation between potential biomarkers, such as *Sneathia and Delftia* found in community state types IV and II, and chronic inflammation with cytokine overexpression. Better analysis of these factors could be of use in the prevention of the progression of the disease and, eventually, in new therapeutic strategies.

## 1. Introduction

Human papillomavirus (HPV) infection is the most frequent sexually transmitted disease, and is the main significant cause of the development of cervical precancerous lesions (dysplasia) and cervical cancer [1,2]. It is estimated that more than 80% of sexually active women contracted this infection at least once in their lifetime, but most cervical HPV infections resolve spontaneously; nevertheless, in a minority of women, the virus persists and progresses to cervical dysplasia and cancer [3]. Despite the number of cases in developed countries significantly reducing, thanks to screening and the increase in HPV vaccinations, cervical cancer is still the fourth most commonly diagnosed cancer worldwide among females [4]. 

There are two peaks of prevalence of the infection: one between 15 and 26 years old, and another one around 45 years old. It is demonstrated that the incidence decreases with age, while persistence increases [5,6]. Among more than 150 HPV genotypes, 10–20 high-risk HPVs (HR-HPV) are strongly associated with cervical cancer development; in particular, HPV16 and HPV18 are the most carcinogenic [7,8].

Known factors associated with a longer persistence of HPV infection and progression to cervical cancer are: a high number of sexual partners, early initiation of sexual activity, the concomitant presence of sexually transmitted infections, impaired immunity, genetic background, use of oral contraceptives, and smoking [9,10,11,12,13]. 

HPV has a particular tropism for the transitional zone of the cervix, which lays between the paved epithelium of the esocervix and the cylindrical epithelium of the endocervix. This zone represents, from a biological standpoint, a vulnerable area because it permits easier access to the basal layer that is a target of some oncogenic agents, such as HPV [14]. The most carcinogenic types of HPV integrate into the cells of the cervical epithelium, determining a process of cellular transformation that leads to the outset of CIN (cervical intraepithelial neoplasia) lesions [15,16].

Recent publications show that immune status is significantly correlated with viral infection, as well as the vaginal microbiota that is involved in the modulation of the immune system of the female lower genital tract [17,18]. Furthermore, some data report a possible association between vaginal microbiota and the development of CIN, suggesting that an abnormal vaginal environment plays a crucial role in the development and progression of CIN [3,5,6,7]. 

Concerning women in their reproductive age, the vaginal normal microbiota is *Lactobacillus*-dominated (LD), with four predominant species: *Lactobacillus crispatus*, *Lactobacillus gasseri*, *Lactobacillus iners, and Lactobacillus jensenii* [19,20]. These species are thought to play key protective roles, by lowering the environmental pH below 4.5 through lactic acid production, by secreting antimicrobial compounds, and through competitive exclusion [20]. Other key species found in the vagina include anaerobes (*Gardnerella*, *Atopobium*, *Mobiluncus*, *Prevotella*, *Streptococcus*, *Staphylococcus*, *Ureaplasma*, *and Megasphaera*) and commensal microorganisms, such as the opportunistic fungus, *Candida albicans* [21].

Ethnicity, genetic disposition, lifestyle and diet, hygiene status, infections, antibiotic use, sexual activity, and estrogen are known to be important risk factors influencing the proper functionality and characteristics of the vaginal microbiome [22,23].

Bacterial vaginosis (BV) represents a dysbiosis of the vaginal microenvironment and microbiome biofilms, correlating with the reduction in *Lactobacillus* spp. and the predominance of microaerophilic bacteria (e.g., *Gardnerella*, *Atopobium*, *Prevotella*, *Sneathia*, *Mycoplasma hominis*, *Haemophilus*) [24,25,26,27]. Recent data from the literature report an association between the dysbiotic non-*Lactobacillus*-dominant (NLD) microbiota of BV and a higher risk of HPV acquisition and its decreased clearance, leading to the development of precancerous dysplasia and progression to cervical cancer [17,24,25,28,29,30,31,32]. In addition, a correlation between increased vaginal pH and the decreased presence of *Lactobacillus* spp. with higher rates of dysbiotic BV in patients with precancerous and cancerous lesions is reported [17,30,31].

The aim of this review is to analyze the interaction between HPV-related cervical intraepithelial neoplasia, vaginal microbiota, and immunological response.

## 2. Results

We found 11 records from the bibliographic search. After excluding three works that were not pertinent with the aim of the research, we accurately examined the other eight papers. We then performed the additional selection, accordingly to the PRISMA diagram, which provided us with a set of six papers. Of the six works under consideration, all investigate the interaction between a *Lactobacillus* non-dominant vaginal microbiota (VMB) and HPV infection; in particular, four of the papers analyze the vaginal community state types (CSTs) and their changes associated with HPV infection. In five studies, microbes correlated with changes in VMB are detected, and in two papers microbe markers of HPV infection/LSIL (low squamous intraepithelial lesion)/HSIL (high squamous intraepithelial lesion)/ICC (invasive cervicale cancer) are identified. Three studies also describe the importance of the increase in vaginal pH in changing VMB, leading to a greater susceptibility to HPV infection. One study attempts to characterize local immune signatures in patients at various stages of cervical carcinogenesis by measuring the levels of some cytokines, assessing that patients with invasive carcinoma, but not precancerous neoplasm, display increased levels of genital immune mediators, leading to chronic inflammation.

Techniques used for HPV detection are commercially available DNA tests, such as the Roche Linear Array HPV Genotyping test, Hybribio HPV typing kit, and Digene Hybrid Capture DNA test, or through an assay of polymerase chain reaction (PCR) using specific primers (Illumina MiSeq).

For the study of vaginal microbiota, the initial assessment includes diagnostic methods such as microscopic evaluation, Gram stain test, and microbiological cultures. In some cases, DNA is extracted through DNA isolating kit.

Among the samples, five out of six are from the vagina, and two of six from the cervix. Only in one case is a double sample taken, one from the vagina and one from the cervix, and the sample from the cervix is diluted. All the other samples are not diluted.

Further characteristics of the studies analyzed are summarized in (Table 1)

## 3. Discussion

### 3.1. Factors Influencing Vaginal Microbiota Composition

The structure of vaginal microbiota (VMB) is controlled by numerous factors. The most intrinsic factor is ethnicity, which is significantly associated with modifications in vaginal composition; Caucasian and Asian women display a greater prevalence of *Lactobacillus*-dominant microbiota, when compared to Hispanic and Black women [20]. These differences are the result of the coaction of genetic factors, which influence the immunity and metabolic pathways, and cultural and social factors, which influence lifestyle (i.e., hygiene practices) [37]. Female hormones also have a great impact on vaginal microbiota, providing the greatest stability over the menstrual cycle at the time of the estrogen peak [38,39], acting in a similar way to hormonal contraceptives; some studies show their association with a lower risk of incidence and lower recurrence and prevalence of BV [40]. Other environmental factors that influence VMB composition are smoking [41] and recent intercourse [42].

### 3.2. Composition of Vaginal Microbiota

Ravel and colleagues [20] sought to develop an accurate comprehension of the composition of the vaginal microbial ecosystem, by characterizing the vaginal microbiota and vaginal pH of 396 asymptomatic, sexually active women, divided equally in four self-reported ethnic groups: White (n = 98), Black (n = 104), Asian (n = 97), and Hispanic (n = 97). They collected two samples from each participant, then the whole genomic DNA was extracted from each swab, and analyzed through the sequencing of barcoded 16S rRNA genes.

Five major groups of microbial communities were retrieved from the analysis, in accordance with previous data in the literature on microbial biofilm specimens of vaginal diversity [43]. The five groups, nominated I, II, III, IV, and V, contain 104, 25, 135, 108, and 21 taxa, respectively. The most heterogeneous group is group IV, and this mirrors the Shannon diversity indices of the microbiome microenvironment.

Unlike other anatomical sites, one or more species of *Lactobacillus,* which constitute >50% of all sequences obtained, represent the most important part of vaginal communities (73%). *Lactobacillus crispatus* represents the main agent of vaginal community I (26.2% of the samples), while *Lactobacillus gasseri* represents the main agent of group II (6.3%), and *Lactobacillus iners* of group III (34.1%), whereas group V (5.3%) is dominated by *Lactobacillus jensenii*. The remaining communities, observed in 27% of the samples, constitute a large heterogeneous group (IV), characterized by higher concentrations of anaerobic bacteria, including *Dialister*, *Prevotella*, *Atopobium*, *Gardnerella*, *Megasphaera*, *Peptoniphilus*, *Sneathia*, *Eggerthella*, *Aerococcus*, *Finegoldia, and Mobiluncus*.

Interestingly, in the communities where the *Lactobacillus* spp. or *Lactobacillus crispatus* (group I) species are represented, a slightly higher pH, between 4.4 (group III) and 5.0 (group II) is exhibited, suggesting that these communities together may not produce levels of lactic acid equal to those reported for group I or, alternatively, they may have different buffering capabilities.

### 3.3. Vaginal Microbiota and Cervical Intraepithelial Neoplasia/Cervical Carcinogenesis and Microbiota Biomarkers

Several studies indicate a correlation between HPV infection and dysbiotic vaginal microbiota. In fact, there is evidence to suggest that persistence of HPV is more probable in those with altered microbiota.

A multicentric cross-sectional study, lead by Laniewski et al., on 100 premenopausal women, shows that, similar to larger epidemiological studies, HPV16 is the most prevalent (65% of HPV-positive samples). Other frequent types identified are HPV45, HPV58, and HPV31. Co-infection with multiple HPV genotypes is uncommon, as well as the decreased severity of cervical dysplasia, consistent with previous data from the literature [44].

As for the vaginal microbiota, Sikao Wu et al. demonstrate that, with the progression of cervical dysplasia, the composition of the vaginal microbiota microenvironment, has corresponding changes. In particular, in the HSIL and CA groups (cervical cancer patients) *Lactobacillus* decreases, while other bacteria, such as anaerobes Prevotella and Megasphaera, increase significantly. Accordingly, in Laniewski et al., patients with advanced cervical dysplasia and invasive carcinoma display a significantly higher rate of diverse non-Lactobacillus-dominant (NLD) vaginal microbiota (VMB) (*p* = 0.04). In particular, *Lactobacillus iners* is reported to increase in HPV-positive women and women with LGD and HGD, being associated with a higher tendency to shift to dysbiotic VMB. In addition, *Lactobacillus iners* may be connected with a higher risk of cervical dysplasia development [45].

Most of the papers analyzed are based on the vaginal microbiome community state types (CSTs) classification (Ravel et al., 2010 [20]); CST I: *Lactobacillus crispatus*-dominated, CST II: *Lactobacillus gasseri*-dominated, CST III: *Lactobacillus iners*-dominated, CST IV: *Lactobacillus*-depleted, and CST V: *Lactobacillus jensenii*-dominated.

In Mitra’s study, by examining the microbiota of 169 women, they identify 5 major clusters that show a bacterial community structure superimposable on previous reported vaginal CSTs, finding that the rate of a CST IV vaginal microbiome doubles in patients with low-grade squamous intraepithelial lesions (LSIL), triples in patients with high-grade squamous intraepithelial lesions (HSIL), and quadruples in patients with invasive cancer. Conversely, the frequency of CST I (*Lactobacillus crispatus*-dominant) is lower with the increase in the severity of the disease. In the LSIL women, a significant over-expression of *Lactobacillus jensenii* (*p* < 0.01) and *Lactobacillus coleohominis* (*p* < 0.05) is observed.

A longitudinal study by Brotman et al. suggests that women with a highly diverse vaginal microbiome, with *Lactobacillus* spp. depletion (CST IV), are predisposed to have a persistent HPV infection, and a long-term positivity to HPV genotyping.

Chen and colleagues also observe that, during HPV infection, there is an increase in vaginal bacterial richness and diversity, despite the status of CINs, causing the shift from vaginal bacterial community structure III to CST IV, and a decrease in the percentage of *Lactobacillus*, even if it remains the most abundant genre in all groups. This results are in agreement with previous studies [24,25,46,47,48].

Conversely, in a study of Mengying Wu et al., cervical intra-epithelial neoplasia (CIN) switches the vaginal bacterial community structure from CST IV to CST II, with the LSIL group dominated by CST I (9/22, 40.9%). Microbiota heterogeneity is found to be more marked in CST II and CST IV (*p* < 0.001), mainly in CST II. When a depletion in *Lactobacillus* is observed in the HSIL group, enrichment in the bacteria associated with bacterial vaginosis is found, even though no statistical connection is demonstrated between CSTs and grades of squamous intra-epithelial neoplasia (*p* = 0.49).

Concerning specific bacterial taxa, Laniewski et al. find novel bacterial genera (i.e., *Shuttleworthia*, *Gemella, and Olsenella*) to have a connection with HPV infection and/or cervical neoplasia (LGD, HGD, or ICC), along with the common bacteria (i.e., *Gardnerella*, *Prevotella*, *Atopobium*, *Megasphaera*, *Parvimonas*, *Anaerococcus*, *Peptostreptococcus*, *Sneathia*), most of which are usually associated with BV. However, they also discover an association, in HPV-positive women, of some bacterial taxa with the lesser-known aerobic vaginitis microbes (AV) (i.e., *Streptococcus agalactiae*), or other dysbioses (i.e., *Clostridium*).

Consistent with the study by Laniewski et al., Plisko and colleagues reveal a relation between AV and cervical dysplastic lesions, showing that AV, but not BV, is found to be significantly more often associated with abnormal pap smear cytology. On the other hand, BV is linked with CIN1, but not with CIN2+ with low-grade lesions being promoted by both HR and LR-HPVs. Furthermore, for the first time, it is observed that AV-associated microbiota changes are also linked to histological pre-invasive cervical lesions, and that there is a positive correlation between the severity of AV and that of the cervical HPV-induced lesions: moderate to severe AV is significantly associated with CIN2+, while BV is not (*p* = 0.013).

Moreover, among the bacteria mentioned above, Laniewski et al. identify *Sneathia* spp. to be the only genus that is significantly increased in patients with HPV infection, or HPV-related cervical dysplastic lesions and cervical cancer. Results of this study complement and extend previous findings [17,25,48] showing *Sneathia*’s correlation with all stages of cervical dysplasia development, suggesting it may be a possible biomarker of CIN progression. On the other hand, Audirac-Chalifour et al. uphold that Sneathia is expressed significantly only in CIN, but not in ICC [17].

Mengying Wu et al. also identify the *Delftia* genus as a potential biomarker of squamous intraepithelial neoplasia. On the other hand, Mitra et al. identify *S. sanguinegens* to be more predominant in women with HGD compared to LGD.

Chen and colleagues observe a negative correlation of HPV infection with the relative richness of *Lactobacillus* spp., *Gardnerella,* and *Atopobium*, and, on the contrary, a positive correlation with the relative richness of *Prevotella*, *Bacillus*, *Anaerococcus*, *Sneathia*, *Megasphaera*, *Streptococcus,*. These changes in the vaginal microbiome are probably related to the progression of the severity of CIN. HPV infection alone, without any cervical lesions, is strongly connected with Megasphaera.

Despite many pieces of evidence suggesting a role of vaginal microbiota in the pathogenesis of HPV-related lesions, some studies still struggle to show a correlation. For example, in Mengying Wu’s study, CSTs and grades of squamous intraepithelial neoplasia have no statistical connection (*p* = 0.49), meaning that there is no significant difference in the vaginal microbiota of women with or without intraepithelial lesions or malignancy. For this reason, they speculate that the more severe the lesion, the greater the vaginal diversity, but this effect is not reported to be statistically significant either, which is in agreement with the results reported by Mitra et al. Their findings suggest that vaginal microbial diversity is associated with the advancing severity of CIN, but does not attain significance. The same results are shown by Huang et al. [49], who show that the diversity of the cervico-vaginal microbiota microenvironment may have no association with the progression of squamous intraepithelial neoplasia.

Above all, we should remember that persistent infection with HR-HPV genotypes represents the most important risk factor for cervical cancer progression and ICC development [50].

### 3.4. Vaginal pH, Lactic Acid and Hydrogen Peroxide

While in other parts of the human body having a diversification of microbes is considered a sign of health, in the vagina, a particularly diverse microbiota is often associated with a state of dysbiosis and disease. Under normal circumstances, *Lactobacillus* is the most dominant bacterial genus in the vagina, and it maintains the stability of the vaginal microenvironment, preventing the colonization of bacterial vaginosis (BV)-associated bacteria, through maintenance of a low pH and the production of bacteriostatic and bactericidal metabolites, such as lactic acid, H_2_O_2_, and biosurfactants and bacteriocins. This is vital for the preservation of the cervical epithelial barrier that inhibits the entry of pathogens, such as HPV, to the basal keratinocytes [51]. Consistent with this, Plisko et al. demonstrate that an increased vaginal pH is more frequent in women with the cervical pathology CIN1+ (53/110, 48.2%) than in women without CIN (30/118, 25.4%, *p* < 0.0001). Other than vaginal microbiota, increased vaginal pH is also correlated with infections, including sexually transmitted ones, and, therefore, could also be associated with sexual activity.

However, there are many other conditions that could modify vaginal pH, such as the use of different vaginal products, hygiene habits (e.g., vaginal douching), low estrogen levels, recent sexual activity, and the presence of sperm or blood. All of these conditions may be taken into consideration.

Recent studies show that CSTs III and IV, characterized by high diversity and low Lactobacilli dominance, are frequently associated with HPV infection and the development of cervical disease [30]. In these groups, when anaerobic bacteria prevail and colonize, they start producing enzymes and metabolites, which may disrupt the epithelial barrier, facilitating the entry of pathogens, such as HPV. They can also activate the cellular pathways responsible for a persistent viral infection, leading to disease development and progression [52].

While the level of H_2_O_2_, lactic acid, and other products is reduced because of fewer producers, and the barrier of cervical and vaginal epithelial mucosa becomes more fragile, the microflora stimulates the production of pro-inflammatory cytokines, leading to the disruption of the epithelial intimal barrier [53].

HPV infection itself is thought to maintain the altered acidic environment of the vagina, and it might lead to changes in the lower genital tract microbiota by inducing host mucosal immune response and genital inflammation. As a result, the vaginal environment is subverted, promoting the abnormal adhesion of HPV in the vagina, causing a local microecological imbalance and destroying the local immune function of the cervix, while simultaneously increasing the adhesion, invasion, and colonization of an abnormal microenvironment. This forms a vicious cycle in lower genital tract tissues, with elevated vaginal pH and non-*Lactobacillus*-dominant VMB leading to chronic inflammation and HPV persistence and a progression in infection, with the possible development of cervical cancer [31,54].

As for pathogenic mechanisms, Plisko et al. also hypothesize that the inflammatory characteristics of AV and HPV-induced cervical dysplastic lesions are crucial for the progression of cervical lesions in invasive cancer. Indeed, one of the critical mechanisms of HPV carcinogenesis is inflammation, shown by increased inflammatory interleukins (IL) in subjects with progressive CIN [55]. Also, inflammation of the uterine cervix is known to cause genotoxic damage through oxidation processes and changes in the cervical microenvironment [56]. Identically, AV is characterized by a different degree of inflammation, with increased vaginal leukocytes, the presence of immune active ‘toxic leukocytes’, and highly increased concentrations of IL-1 and IL-6.

### 3.5. Role of Cytokines and Inflammation Response in Cervical Intraepithelial Neoplasia

In VMB there is a complex balance between microorganisms. *Lactobacillus* competes with pathogens, and inhibits their growth by secreting lactic acid, bacteriocins, and H_2_O_2_ and, furthermore, by activating the complement system, triggering a local immune response. As mentioned earlier, when pathogens colonize the vaginal epithelium, this leads to a waterfall effect with an elevated pH and the adhesion of pathogens, such as HPV, to the epithelium, causing disruption and the activation of a host immune response with chronic genital inflammation.

Many studies investigate the role of the inflammatory response in patients with cervical intraepithelial neoplasia or cervical cancer. In most of them, a chronic inflammatory response with the deregulation of pathways and the overexpression of cytokines, is found only in CIN2-3 or ICC. In accordance with this, the HR-HPVs, which are proven to be the main motive of cervical cancer, down-regulate cellular tumor suppressors (pRB and p53) and epidermal growth factor receptors (EGFR), through HPV oncoproteins (E5, E6, and E7) integrating into the cellular DNA and stimulating some inflammation pathways. E6 and E7 are also demonstrated to have an association with elevated levels of NF-κB [7].

Laniewski et al. evaluate the expression of specific mediators of the immune system in cervico-vaginal washings; the authors report that pro-inflammatory and chemotactic cytokines such as IL-36γ, TNFα, RANTES, MIP-1α, MIP-1β, RANTES, and IP-10; hematopoietic cytokines, as well as Flt-3L and GM-CSF; cytokines of adaptive immunity such as IL-2, IL-4, and sCD40L; and an anti-inflammatory cytokine IL-10, are significantly expressed in ICC. The authors then underline that the genital inflammatory pathway is significantly activated in ICC patients (60%) compared to HPV− patients (5.0%; *p* = 0.02), HPV+ subjects (6.5%; *p* = 0.01), and women with high-grade cervical dysplasia (3.7%; *p* = 0.006). ICC, but not low-grade or high-grade cervical intraepithelial neoplasia, shows a significant increase in the inflammatory score system contrary to HPV− subjects (*p* < 0.05) [31]. As for the connection between VMB and immune system mediators, they notice that all those tested are associated with the presence of Sneathia,, suggesting its possible role in the inflammatory response.

Likewise, Audirac et al. report that some agents of the cervical microbiota communities may cause a modification of cytokine expression in the cervical microenvironment, especially during the progression of CIN into ICC. *Sneathia* and *Fusobacterium* spp. either contribute to shift Th1 immunity to Th2, increasing the levels of IL-4 and TGF-β1, or directly intervene in the E-cadherin/β-catenin signaling pathway in cervical HPV-infected cells, leading to the activation of the transcription factor NF-kB in the nuclei [17].

Consistent with previous studies, there is evidence that HPV-associated cervical lesions (Figure 1) show a change in cytokine patterns, meaning that a discrepancy in the proportion of T helper 1 (Th1)/Treg, with a higher level of T reg, gives rise to facilitated tumor progression and immunity evasion, leading to ICC. Such changes are manifested as a reduction in IL-2, IL-12, and TNF-α, and an increment in transforming growth factor-beta (TGF-β), IL-6, IL-8, TNF-α, macrophage inflammatory protein 1 alpha (MIP-1α), granulocyte-macrophage colony-stimulating factor (GM-CSF), IL-1α, and IL-1β [57,58].

## 4. Materials and Methods

In order to perform a literature review, we followed the preferred reporting items for systematic reviews and meta-analyses (PRISMA) guidelines [59]. The medical literature on the role of vaginal microbiota in cervical intraepithelial neoplasia available between November 2015 and September 2021 was reviewed, screening the Medical Literature Analysis and Retrieval System Online (PubMed/MEDLINE) and the Cochrane Central Register of Controlled Trials (CENTRAL). The search strategy used in both databases included the combination of key words and MeSH terms “microbiota” AND “cervical intraepithelial neoplasia” OR “cervical dysplasia”. From 11 records screened, 6 were recognized as eligible for this review. Details are provided in Figure 2.

Two authors (M.F.P. and M.L.V.) independently researched reference lists of recognized manuscripts, aiming to perform an integration of the literature. Interventional, observational, prospective, and retrospective studies were considered. Exclusion criteria were the following: single case reports, book chapters, and conference abstracts. In addition, publications written in a language other than English, and in vitro or in animals experimental trials were excluded. The bibliographic research provided a preliminary group of papers, which were carefully analyzed. Two investigators (M.D. and B.G.) independently evaluated the risk of bias to confirm the validity of the selected papers and overcome possible selection, performance, detection, attrition, and reporting bias, according to the Cochrane Handbook for Systematic Reviews of Interventions [60,61,62]. For the specific purposes of the present review, we then performed a further selection of the preliminary set of papers, with more restrictive criteria. For this reason we analyzed a restricted group of articles in our review, as we reported in the results section.

## 5. Conclusions

Recent studies in the literature suggests that vaginal microbiota may play a possible role in the onset and progression of cervical intraepithelial neoplasia, in particular high-grade CIN, even if, for some authors, this correlation does not attain significance. Therefore, what is yet to be examined is the interaction between all of the factors that are involved in this relationship, in order examine if it is significantly relevant or not.

First of all, the insurgence of CIN, and its progression in high-grade lesions, inevitably requires the presence of previous HPV integration in the cervical tissue, and the activation of oncogenic protein expression. In addition, data in the literature underline that HR-HPVs are the main etiological factors involved in the development and progression of CIN and cervical cancer. Instead, the motivation of why only in some cases HPV infection is cancerous, whereas in other women it is suppressed, may be explained by immunological response system diversity. After infection, in fact, there are many factors involved in the acquisition and persistence of HPV leading to the development of cervical lesions, and immunological response may probably be more significantly responsible in the clearance of HPV infection.

Among extrinsic factors to consider are hormonal contraceptives, recent intercourse, hygiene practices, and smoking, while, among intrinsic factors, the most important is ethnicity, which may also mirror in different genetic settings and in the composition of vaginal microbiota biofilms, resulting in the insurgence of vaginal dysbiosis, as BV and the lesser-known AV, which seem correlated with HPV infection and persistence. In consideration of this, additional biological analysis is required to confirm the role of the vaginal microbiota in the pathogenesis of CIN and cervical cancer. In addition, it is important to investigate the cause–effect relationship between them, in order to understand if the changes in the microbiota microenvironment are the cause, or the consequence, of HPV infection.

Furthermore, it could be crucial to find potential biomarkers, such as Sneathia and *Delftia*, that may be used as predictors of the evolution of HPV lesions, and to investigate the pathogenesis of chronic inflammation that leads to cervical cancer. It is already known, in fact, that the inflammatory response is associated with CIN2+ lesions. Finding predisposing factors in HPV+/CIN1 lesions may be of use in preventing the progression of the disease.

The information about vaginal microbial composition, biomarkers, and chronic inflammation may grant the opportunity for the development of new therapeutic strategies for CIN.

## Figures and Tables

**Figure 1 ijms-23-07174-f001:**
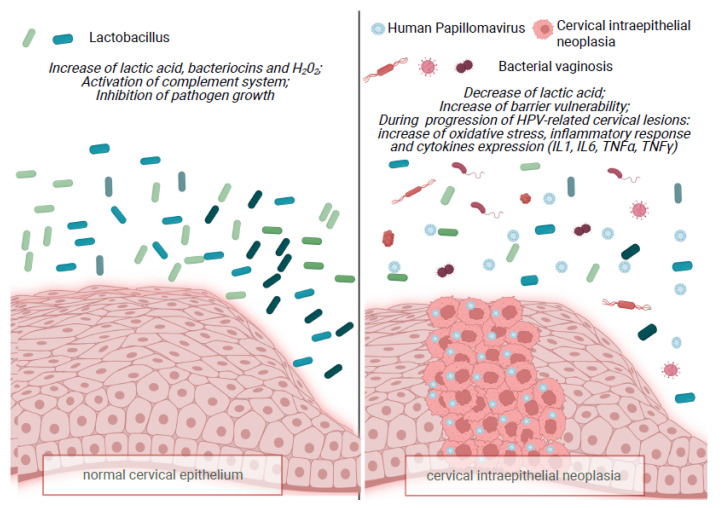
Summary of the interplay of factors influencing vaginal microbiota. Created with Biorender.com.

**Figure 2 ijms-23-07174-f002:**
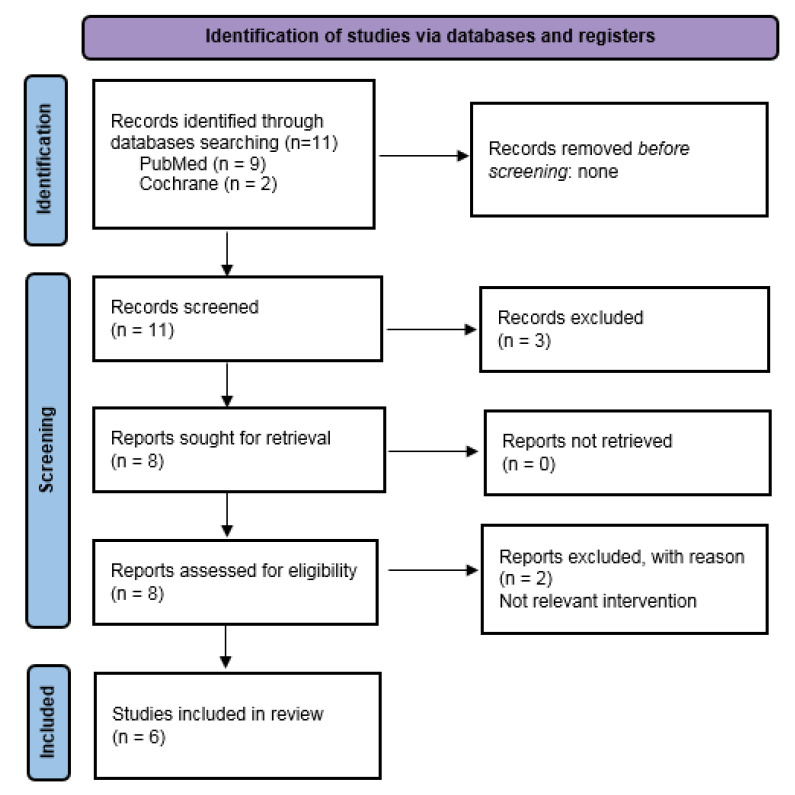
PRISMA 2009 flow diagram with the research strategy.

**Table 1 ijms-23-07174-t001:** Characteristics of studies exploring the association of HPV infection and cervical preinvasive disease to the vaginal microbiome.

Authors	Country	Population Age, Study Size and Design	Microbial Sampling	HPV Detection and Groups	Summary Findings (Risk and Protective Factors, Vaginal pH)
Plisko et al., 2021 [33]	Riga, Latvia; East Clinical University Hospital Outpatient department	112 patients, 19–59 years, prospective case–control study	Scraping from upper vaginal fornix Wet-mount microscopy	31 CIN1, 57 CIN2, 21 CIN3, 1 ICC	− Abnormal vaginal microbiota and msAV are more associated with CIN2+ compared to CIN1 (*p* = 0.013) − BV is found, especially in CIN1 patients (8/31, 25.8%, *p* = 0.009) − Vaginal pH increases more in CIN1+ (53/110, 48.2%) than in negative samples (30/118, 25.4%, *p* < 0.0001).
Mitra et al., 2015 [30]	London, UK; colposcopy and gynecology clinics at Imperial College NHS Healthcare Trust	169 patients, 18–45 years, prospective case–control study	Scraping from posterior vaginal fornix BBLTM Culture SwabTM containing liquid Amies (Becton Dickinson, Oxford, UK). Genomic bacterial DNA extracted using a QiAmp Mini DNA kit (Qiagen, Venlo, The Netherlands)	20 normal, 52 LSIL, 92 HSIL, 5 ICC llumina MiSeq sequencing of 16S rRNA gene amplicons	− CST * IV is associated with increasing dysplasia severity (10 % normal; 21 % LSIL; 27 % HSIL; 40 % ICC) − CST I is negatively associated with increasing disease severity (50 % normal; 42 % LSIL; 40 % HSIL; 20 % ICC) − Higher levels of *Sneathia sanguinegens* (*p* < 0.01), *Anaerococcus tetradius* (*p* < 0.05), and *Peptostreptococcus anaerobius* (*p* < 0.05) are more associated with HSIL than with LSIL − Lower levels of *Lactobacillus jensenii* (*p* < 0.01) are more associated with HSIL than with LSIL − CST IV (higher presence of *Atopobium* spp.) has slower regression of HPV, whereas CST II (higher presence of *Lactobacillus gasseri*) is associated with the most rapid regression rates for HPV
Mengying Wu et al., 2020 [34]	Shanghai, China; Obstetrics and Gynecology Hospital of Fudan University	69 premenopausal, non-pregnant patients, prospective case–control study	Scraping from posterior vaginal fornix Deep sequencing of bar-coded 16S rRNA gene fragments (V3–4) using Illumina MiSeq	31 normal, 22 LSIL, 16 HSIL	− CST II has lower level of *Lactobacillus* spp. and higher level of strictly anaerobic organisms (*Gardnerella*, *Atopobium*, and *Prevotella*) − *Prevotella* and *Streptococcus* are increased in HSIL detection − CIN converts the vaginal bacterial community structure from CSTs IV to II − Microbiota diversity is more marked in CST types II and IV (*p* < 0.001), above all in type II − Enrichment in the *Delftia* genus is found in the LSIL and HSIL groups
Laniewski et al., 2018 [31]	Phoenix (AZ), USA; St. Joseph’s Hospital and Medical Center (SJHMC), University of Arizona (UA) Cancer Center, Maricopa Integrated Health System (MIHS)	100 premenopausal patients, multicentric cross-sectional study	First swab scraping from lateral walls vagina using eSwab collection system containing Amies transport medium (COPAN diagnostics, Murrieta, CA) Second swab with cervico-vaginal lavages (CVL) collected using 10 mL of sterile saline solution 0.9% Both swabs analyzed with PowerSoil DNA Isolation Kit	20 HPV−, 31 HPV +, 12 LGD, 27 HGD, 10 ICC -Linear Array HPV Genotyping Tests (Roche, Indianapolis, IN)	− Control group (HPV−) has the lowest quantity of abnormal pH ** (55.0%, 11/20). Number of patients with abnormal pH differs significantly among groups (*p* = 0.006), and increases with the progression of the disease: 77.8% (21/27) in Ctrl HPV+, 72.7% (8/11) in LGD, 92.6% (25/27) in HGD, and 100% (9/9) in ICC − Only 1.3% (7/79) women are infected with LR HP types. The most represented HPV types in this cohort are HPV16 (64.6%, 51/79), HPV45 (21.5%, 21/79), HPV58 (20.3%, 16/79), and HPV31 (18.9%, 15/79) − Rates of *Lactobacillus*-dominant (LD) VMB (defined as ≥80% relative abundance of Lactobacillus spp.) are significantly decreased, however, dysbiotic non-*Lactobacillus*-dominant (NLD) VMB increases in LGD (67%), HGD (56%), and ICC (80%); on the contrary, NLD VMB decreases in HPV− (40%) and HPV+ (32%) patients (*p* = 0.04) − Higher levels of *Sneathia* spp. in all groups − *Lactobacillus iners* higher in HPV-positive women, LGD, and HGD
Sikao Wu et al., 2021 [35]	Nanning, China; Hospital of Guangxi Medical University	94 patients, 18–52 years, prospective case–control study	Cervix mouth PowerMax (stool/soil) DNA isolation kit (MoBio Laboratories, Carlsbad, CA, USA) 16S rRNA gene sequences. V4 region amplificated by PCR, using the primers 515F and 806R	28 normal, 12 HPV +, 10 LSIL, 31 HSIL, 13 ICC Hybribio HPV typing kit (Chaozhou Hybribio Biotechnology Co., Ltd.) for PCR, and membrane hybridization. 2.2.	− Diversity of vaginal microbiota is higher in more severe cervical intraepithelial neoplasia (*p* < 0.05) − CST III (represented by *Lactobacillus. Iners*, more than 50%) is associated with LSIL (n = 7/10), NH (n = 6/12), NN (n = 12/28) − CST IV is associated with ICC (n = 9/13) and HSIL (n = 14/31). CST IV is a heterogeneous group characterized by reduction in *Lactobacillus* spp (less than 50%), with a rising of strictly anaerobic species (*Gardnerella*, *Megasphera*, *Sneathia*, and *Prevotella*) − Marker microbes are identified for each group: LSIL (*Lactobacillus*, *Xanthobacter*, *Thermus*, *Flavisolibacter*, *Sphingopyxis*, *Sediminibacterium*, *Geobacillus*); HSIL (*Sneathia*); ICC (*Prevotella*, *Mycoplasma*, *Porphyromonas*, *Megasphaera*, *Campylobacter*, *Dialister*, *Peptoniphilus*, *Peptostreptococcus*, *Anaerococcus*)
Chen et al., 2020 [36]	Shanghai, China; Department of Gynecology and Obstetrics, Ren ji Hospital, School of Medicine, Shanghai Jiao Tong University	229 patients, 25–69 years, cross-sectional study	Scraping from the lateral and posterior vaginal fornix 16S rRNA gene sequences. V3 and V4 region amplified by PCR using the Primers 338F and 806R	68 normal, 51 LSIL, 23 HSIL, 9 ICC Deep sequencing of barcoded 16S rRNA gene fragments (V3–4) using Illumina MiSeq	− CST III is mostly associated with normal samples − HPV infection converts vaginal bacterial community structure from CST III to CST IV, increasing vaginal bacterial richness and diversity − *Lactobacillus* spp. is the most abundant genus in all groups − HPV infection reduces the abundance of *Lactobacillus*, *Gardnerella*, *Atopobium*, and, conversely, reduces the relative abundance of *Prevotella*, *Bacillus*, *Anaerococcus*, *Megasphaera*, *Anaerococcus sneathia*, and *Streptococcus* − *Prevotella amnii* is the prevalent bacterium in the LSIL group − Vaginal microbiota with *Lactobacillus* depletion, elevated vaginal pH, and genital chronic inflammation are associated with HPV persistence and progression

* CSTs = community state types. There are 5 major types, each dominated mainly by a specific type of microbe: CST I *Lactobacillus crispatus*; CST II *Lactobacillus gasseri*; CST III *Lactobacillus iners*; CST V *Lactobacillus jensenii*; CST IV is characterized by low numbers of *Lactobacillus* spp. with the increase in diversity of anaerobic bacteria. ** Abnormal vaginal pH is defined as >4.5 and normal vaginal pH is defined as ≤4.5. LGD = low-grade disease, HGD = high-grade disease, ICC = invasive cervical cancer, VMB = vaginal microbiome, msAV = moderate to severe aerobic vaginitis.

## Data Availability

Not applicable.

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
