# Peer review of "The Complex Interplay between Vaginal Microbiota, HPV Infection, and Immunological Microenvironment in Cervical Intraepithelial Neoplasia: A Literature Review"

_ijms, 2022, doi:10.3390/ijms23137174_

Round 1

Reviewer 1 Report

Here are my comments:

[1] Please read through the manuscript carefully for abbreviations which have not been explained. For example, in the abstract the abbreviation CST is used without spelling out what it stands for.

[2] It would be helpful to have a table of some sort with all the abbreviations listed and what they stand for. It would make for an easier read.

[3] Perhaps the style for stating nomenclature has changed, but I thought that, for example, bacterial nomenclature is always italicized. If so, the microorganisms described in the manuscript need to be edited. Please check instructions for the journal as well.

Author Response

1.Please read through the manuscript carefully for abbreviations which have not been explained. For example, in the abstract the abbreviation CST is used without spelling out what it stands for.

A correction was made

2.It would be helpful to have a table of some sort with all the abbreviations listed and what they stand for. It would make for an easier read.

A table with abbreviation was prepared in according with the review suggestion.

3.Perhaps the style for stating nomenclature has changed, but I thought that, for example, bacterial nomenclature is always italicized. If so, the microorganisms described in the manuscript need to be edited. Please check instructions for the journal as well.

The corrections were made in according with the review suggestions

Reviewer 2 Report

Dr. Gardella and colleagues conducted a systematic review on a very interesting topic. This article tries to put order on an aspect not yet fully understood. Infact, persistent HPV infection can progress or spontaneously clear and vaginal microbiota and immune response are the two factors often cited in the literature as conditions to shift in one or the other direction.

The article is clear and data are presented correctly.

English language and style could be improved

I have detected these small errors:

-Line 55: the acronym CIN has not yet been presented. We need to move the explanation from line 61

-Line 82: Bacterial vaginosis is always contract in BV (here VB)

Line 113: please explain here or in discussion section why you consider these three articles irrelevant (redundant?)

Figure 1: **next to record excluded have no explanation in the caption

Table1: no Title is present for table

Table 1: For Mengying Wu et al. article, no informationabout age patients or menopausal state is present; this is very important because the vaginal microbiota changes for ethinia and a lot during the woman's life

Table1: For Laniiewski et al., section please check amounth (typo?)

Line 227: please check genra (did you mean genera?)

Author Response

1.English language and style could be improved

English editing was performed.

2.Line 55: the acronym CIN has not yet been presented. We need to move the explanation from line 61

The correction was made

3.Line 82: Bacterial vaginosis is always contract in BV (here VB)

The correction was  made

4.Line 113: please explain here or in discussion section why you consider these three articles irrelevant (redundant?)

An explanation was included

5.Figure 1: **next to record excluded have no explanation in the caption

The correction was made

6.Table1: no Title is present for table

The title was added in the manuscript

7.Table 1: For Mengying Wu et al. article, no informationabout age patients or menopausal state is present; this is very important because the vaginal microbiota changes for ethinia and a lot during the woman's life

The corrections were made

8.Table1: For Laniiewski et al., section please check amounth (typo?)

The correction was made

9.Line 227: please check genra (did you mean genera?)

 The correction was made.

Round 2

Reviewer 1 Report

Please go through the manuscript and spell out all the names of the microorganisms you are discussing. For example, L. iners should be spelled out as Lactobacillus iners. This applies to all instances throughout.

[1] In the abstract:

-Please spell out CST

-Please italicize : Lactobacillus spp, Sneathia and Delfti

[2] Table 1:

The following should be spelled out

S. sanguinegens

A. tetradius

P. anaerobius

L. iners (appears twice)

[3] page 8

Needs to be spelled out

S. sanguinegens

Please check throughout manuscript.

Author Response

The name of the microorganisms were be spelled out in  all sections of manuscript  and in the table, as  it was requested.

In the abstract

- CST was be spelled out.

 - the name of Lactobacillus spp, Sneathia and Delfti were italicized.